# Optimization of Microwave-Assisted Extraction Process of Total Flavonoids from *Salicornia bigelovii* Torr. and Its Hepatoprotective Effect on Alcoholic Liver Injury Mice

**DOI:** 10.3390/foods13050647

**Published:** 2024-02-21

**Authors:** Dujun Wang, Jing Lv, Yan Fu, Yueling Shang, Jinbin Liu, Yongmei Lyu, Ming Wei, Xiaohong Yu

**Affiliations:** School of Marine and Bioengineering, Yancheng Institute of Technology, Yancheng 224051, China; wangdj@ycit.edu.cn (D.W.); 19802687360@163.com (J.L.); 13382422508@163.com (Y.F.); ylshang33@126.com (Y.S.); jinbin8810@ycit.edu.cn (J.L.); lyu.yongmei@ycit.edu.cn (Y.L.); weiming718@163.com (M.W.)

**Keywords:** *Salicornia bigelovii* Torr., ultrasound-assisted extraction, response surface methodology, flavonoids, hepatoprotective effect

## Abstract

The objective of this study was to determine the optimal extraction conditions for total flavonoids from *S. bigelovii* using microwave-assisted extraction and to analyze the protective effect of total flavonoids from *S. bigelovii* on alcoholic liver injury in mice. The optimization of the process conditions for the microwave-assisted extraction of total flavonoids from *S. bigelovii* was performed using response surface methodology, and an alcohol-induced acute liver injury model in mice was used to investigate the effects of different doses of total flavonoids (100 mg/kg, 200 mg/kg, and 400 mg/kg) on the levels and activities of serum alanine aminotransferase kits (ALT), glutamic oxaloacetic transaminase kits (AST), superoxide dismutase kits (SOD), glutathione peroxidase kits (GSH-Px), and malondialdehyde (MDA). We performed hematoxylin–eosin (H&E) staining analysis on pathological sections of mouse liver tissue, and qRT-PCR technology was used to detect the expression levels of the inflammatory factors *IL*-1 β, *IL*-6, and *TNF*-α. The results revealed that the optimal extraction process conditions for total flavonoids in *S. bigelovii* were a material-to-liquid ratio of 1:30 (g/mL), an ethanol concentration of 60%, an extraction temperature of 50 °C, an ultrasound power of 250 W, and a yield of 5.71 ± 0.28 mg/g. Previous studies have demonstrated that the flavonoids of *S. bigelovii* can significantly inhibit the levels of ALT and AST in the serum (*p* < 0.001), reduce MDA levels (*p* < 0.001), increase the activity of the antioxidant enzymes SOD and GSH-Px (*p* < 0.001), and inhibit the *IL*-1 β, *IL*-6, and *TNF*-α gene expression levels (*p* < 0.001) of inflammatory factors. The total flavonoids of *S. bigelovii* exert a protective effect against alcoholic liver injury by reducing the levels of inflammation, oxidative stress, and lipid peroxidation caused by alcohol. The results of this study lay the foundation for the high-value utilization of *S. bigelovii* and provide new resources for the development of liver-protective drugs.

## 1. Introduction

*Salicornia bigelovii* Torr. (*S. bigelovii*) is an annual herbaceous plant belonging to the family Chenopodiaceae. Its seeds germinate directly in seawater, thus making it one of the most salt-tolerant plant species in the world [1]. *S. bigelovii* is a plant that combines dietary and medicinal uses. It is consumed as a common vegetable or as a medicinal herb. Research has demonstrated that as a seasonal sea vegetable, *S. bigelovii* possesses a curious, salty, and crisp taste and is rich in protein, various trace elements, cellulose, vitamins, and essential amino acids [2]. Therefore, *S. bigelovii* could be consumed as a green organic vegetable. According to the “Chinese Materia Medica”, *S. bigelovii* exhibits protective effects against various diseases such as reducing blood pressure, diuresis, and swelling and treating headaches and liver yang hyperactivity caused by yin deficiency. Individuals in Xinjiang often use *S. bigelovii* as a medicinal herb to treat sepsis. In South Korea and India, *S. bigelovii* is often used to treat obesity, diabetes, scabies, gastrointestinal imbalances, and other diseases. Research has demonstrated that various active ingredients such as flavonoids, alkaloids, polysaccharides, and saponins can be isolated from *S. bigelovii* and exert anti-inflammatory [3], antioxidant [4], antitumor [5], hypoglycemic, and lipid-lowering effects [6]. Research has confirmed that *S. bigelovii* possesses a high content of flavonoids [7], alkaloids, chromones [8], and sterols [9], and Quercetin and Quercetin-3-O have also been isolated from *S. bigelovii* along with β-D-glucoside, Rutin, and Isorhamnol-3-O-β-flavonoids such as D-glucoside. Previous studies have determined that *S. bigelovii* contains multiple flavonoids, and camellianin A and noreugenin, which are previously unreported compounds, have been identified [10].

The traditional extraction methods for flavonoids include solvent extraction, crystallization, microfluidic technology, vacuum extraction, Soxhlet extraction, and so on. However, these methods have problems such as long extraction time, expensive extraction equipment, low extraction efficiency, and the need for high-purity solvents. The ultrasonic assisted extraction technology is based on the presence, polarity, and solubility of other active ingredients in the substance, which quickly enter the solvent under the action of ultrasonic waves to obtain a multi-component mixture, and then to obtain the active substance monomer through appropriate separation and purification techniques. At present, this method has been widely used for the extraction and separation of effective components in natural products. Combining ultrasonic waves with traditional solvent extraction has the advantages of reducing extraction time, targeted heating, reducing solvent consumption, and high extraction rate, making it an effective method for extracting total flavonoids [11]. It has been used to extract various plant flavonoids [12,13]. Research has found that using ultrasound-assisted extraction of total flavonoids can shorten the extraction time and improve the yield compared to those of traditional Soxhlet extraction methods [14]. Ultrasound-assisted extraction avoids the damage of high temperature to active ingredients, and the main influencing factors of ultrasound extraction include ultrasound frequency, extraction time, etc. Therefore, finding the appropriate parameters in ultrasound-assisted extraction is the key to improving extraction efficiency. Meanwhile, the combination of ultrasound technology and other emerging technologies will be a research hotspot. With the continuous updating of research technology, ultrasonic extraction technology will show more extensive application prospects in fields such as food, medicine, and chemical industry [15].

Plant flavonoids are an important class of compounds possessing various physiological activities. Studies have indicated that flavonoids exert significant therapeutic effects against liver damage caused by drugs, alcohol, and other chemicals [16,17]. The study showed that total flavonoids could reduce the mRNA expression levels of peroxisome proliferator-activated receptors (PPARs), a key factor in fat metabolism in fatty liver tissue, and this is a potential target for the treatment of steatohepatitis [18]. Studies have shown that total flavonoids can reduce the inflammatory factors NF-κB and TNF-α in rat liver tissue to reduce the damage to rat liver. Studies have revealed that puerarin treatment can alleviate liver necrosis and vacuolization due to its role in activating the adenosine 5′-monophosphate (AMP)-activated protein kinase (AMPK) pathway, clearing free radicals, and inhibiting lipid peroxidation, inflammatory mediators, and the caspase-3 pathway [19].

In recent years, there has been an increasing amount of research examining *S. bigelovii*, with most studies focusing on salt tolerance. However, there are no reports detailing the protective effects of the total flavonoids from *S. bigelovii* against alcoholic liver injury. To further study and develop the utilization of *S. bigelovii*, we used the yield of total flavonoids from *S. bigelovii* as an indicator and performed response surface analysis to determine the optimal extraction conditions for the microwave-assisted extraction of total flavonoids from *S. bigelovii*. The hepatoprotective effects of the total flavonoids from *S. bigelovii* were analyzed using a mouse alcoholic liver injury model, thus laying the foundation for the high-value utilization of *S. bigelovii* and providing new resources for the development of hepatoprotective drugs.

## 2. Materials and Methods

### 2.1. Materials and Instruments

Fresh *S. bigelovii* (batch number: 20210610) was purchased from Yancheng Lvyuan Salt Soil Agriculture Co., Ltd. (Yancheng, Jiangsu, China). After identification by Professor Yu Xiaohong from the Yancheng Institute of Technology (Yancheng, Jiangsu, China), the sample was classified as the whole plant of *S. bigelovii* belonging to the *Salicornia bigelovii* Torr. Anhydrous ethanol, sodium hydroxide, anhydrous ether, sodium nitrite, and aluminum nitrate were of analytically pure quality. Rutin was purchased from Shanghai Yuanye Biotechnology Co., Ltd. (Shanghai, China), and alcohol (55 degree, Beijing Hongxing Brewing Co., Ltd., Beijing, China), silymarin capsules (Tianjin Tianshili Shengte Pharmaceutical Co., Ltd., Tianjin, China, batch number 050,707,041), AST (batch number 20,201,015), ALT (batch number 20,201,016), SOD (batch number 20,201,013), GSH-Px (batch number 20,201,015), and MDA (batch number 20,201,015) were all purchased from Nanjing Jiancheng Biotechnology Research Institute (Nanjing, China). The Revert Aid First Strand cDNA Synthesis Kit was purchased from Thermo Fisher Scientific (Waltham, MA, USA), and the fluorescence quantification kit was purchased from Nanjing Novozan Biotechnology Co., Ltd. (Taq Pro Universal SYBR qPCR Master Mix, Nanjing, China). A fluorescence quantitative PCR instrument (LC-96, Roche, Switzerland), UV2100 UV spectrophotometer, XO-SM50 ultrasonic combination reaction system, and H1850R desktop high-speed freezing centrifuge were used for the experiments.

### 2.2. Experimental Animals and Grouping

Specific pathogen-free (SPF) grade 8-week-old Kunming male mice (48) with a body weight of 20 ± 2 g were purchased from Charles River Experimental Animal Technology Co., Ltd. (Beijing, China), with the animal qualification certificate number of SCXK (Beijing) 2016-0011. After one week of adaptive feeding, the mice were used for the experiment. They were maintained at a room temperature of 22–25 °C, a relative humidity of 65%, alternating light and dark illumination for 12 h, and allowed to eat and drink freely. Kunming mice were randomly divided into a blank control group, a model group, a positive control group (silymarin 200 mg/kg), a low-dose *S. bigelovii* group (100 mg/kg), a medium-dose *S. bigelovii* group (200 mg/kg), and a high-dose *S. bigelovii* group (400 mg/kg) with 8 mice in each group. The blank control and model groups were administered equal volumes of physiological saline by gavage for 4 consecutive weeks. After the last administration, a mouse model of acute alcoholic liver injury was induced by gavage of 6 g/kg alcohol with alcohol administered every 12 h for a total of 3 times [20]. All the animal experimental procedures were approved by the Animal Ethics Committee of Jiangsu Vocational College of Medicine (approved identification: XMLL-2022-840).

### 2.3. Samples and Processing of S. bigelovii

Fresh *S. bigelovii* plants were dried and crushed using a grinder. The *S. bigelovii* powder was soaked in anhydrous ether for degreasing, and the supernatant was discarded. The lower layer was removed and allowed to evaporate in a fume hood. It was then dried in a 60 °C oven, sealed in a sealing bag, and stored in a cool and dry place for later use.

### 2.4. Creation of Standard Curve for Rutin

A 0.5 mg amount of rutin standard solution was dissolved in 60% ethanol, transferred to a brown 50 mL volumetric flask, and brought to a volume of 60% ethanol to obtain a rutin standard solution with a mass concentration of 0.1 mg/mL. Rutin standard solutions (0, 1, 2, 3, 4, and 5 mL) were placed in a test tube. First, 0.3 mL of 5% sodium nitrite was added, and the mixture was shaken for 6 min. Next, 0.3 mL of 10% aluminum nitrate solution was added, and the mixture was shaken for 6 min. Finally, 4.0 mL of 5% sodium hydroxide and 95% ethanol were added to a total volume of 10 mL. The absorbance was measured at OD_510_ [21]. A standard curve was drawn with the rutin concentration (mg/mL) as the *x*-axis and absorbance as the *y*-axis.

### 2.5. Extraction and Content Determination of Total Flavonoids from S. bigelovii

Utilizing the method of Huang et al. [22] with slight modifications, we accurately weighed defatted sea cucumbers (0.5 g) in a 50 mL conical flask with a stopper. A 60% ethanol solution was added at a solid–liquid ratio of 1:60 and stirred evenly. We set the temperature to 60 °C and the ultrasound power to 250 W, and we extracted the samples for 40 min under these conditions. After the extraction was completed, the samples were transferred to a centrifuge tube and centrifuged for 4 min at 8000 rpm. The supernatant obtained by centrifugation contained the flavonoids of *S. bigelovii*. The crude extract (2 mL) was transferred to a 10 mL volumetric flask. First, we added 0.3 mL of 5% sodium nitrite, and the mixture was shaken and incubated for 6 min. Next, 0.3 mL of 10% aluminum nitrate solution was added, and the mixture was shaken for 6 min. Finally, 4.0 mL of 5% sodium hydroxide and 95% ethanol were added to a total volume of 10 mL. The plates were shaken thoroughly for 15 min. The rutin standard was used as a blank control to measure the absorbance at 510 nm and to calculate the yield of total flavonoids in *S. bigelovii* using the following formula:W = (C × V × D)/M
where W is the yield of flavonoids from sea chestnuts (%), C is the concentration of flavonoids in the test solution (g/mL), D is the dilution factor, V is the volume of the extraction solution (mL), and M is the sample mass (g).

### 2.6. Response Surface Experimental Design

Based on the results of the single-factor experiment, four primary influencing factors were determined, and they included the solid–liquid ratio (A), the ethanol volume fraction (B), the extraction temperature (C), and the extraction power (D). Three levels of low, medium, and high were designed for each factor and were represented by −1, 0, and 1, respectively. The Box–Behnken design model from Design Expert 10.0.4 software was utilized, and the total flavonoid yield was used as the evaluation index (Y). A response surface experiment was designed using the four factors and three levels listed in Table 1. The number of independent experiments was based on a single factor experiment, and according to the Box–Behnken central combination design principle, a total of 29 independent experiments were conducted, listed Table 2, of which 24 were factorial experiments and 5 were central experiments, which were used to evaluate prediction errors. 

### 2.7. The Protective Effect of Total Flavonoids from S. bigelovii on Alcoholic Liver Injury

#### 2.7.1. Determination of Biochemical Indicators of ALT, AST, SOD, GSH-Px, and MDA in Mouse Serum

At the end of the last alcohol gavage, the mice were euthanized by CO_2_ asphyxiation, blood was collected from the heart and placed in a centrifuge tube with heparin sodium at room temperature for 30 min, the tubes were centrifuged at 900× *g* for 10 min, the supernatant was collected, and the reagent kit was used according to the kit instructions to detect the levels of ALT, AST, SOD, GSH-Px, and MDA in the serum of each group of mice.

#### 2.7.2. Analysis of Organ Coefficients and Liver Pathological Tissue Slices in Mice

Each group of mice was weighed before euthanasia and immediately euthanized after blood collection. Organs such as the liver were separated, and their surface blood was washed with physiological saline. After absorption using absorbent paper, the weights of the organs were accurately measured. We acquired the right lobe liver tissue of mice, washed the surface blood with physiological saline, absorbed the surface water with absorbent paper, soaked the liver in a 10% formaldehyde solution for dehydration and fixation, embedded it in paraffin, and sliced it to a thickness of 5 μm. Then, we performed H&E staining and observed the morphology of liver tissue cells under a light microscope. The Ishak scoring system was used to evaluate the degree of pathological changes in liver tissue. The main evaluation indicators include liver cell necrosis, fibrosis, inflammation, and biliary tract injury [23]. According to the degree of liver cell necrosis, it was divided into no necrosis, less than 50% necrosis, 50–75% necrosis, and more than 75% necrosis, with a score of 0–3 points set in sequence. According to the degree of fibrosis, it could be divided into the following: no fibrosis, only the presence of fibrosis phenomenon but not severe, mild fibrosis with reticular fibrosis and interlobular fibrosis, moderate fibrosis with obvious coarse fibrosis in reticular and interlobular fibrosis, severe fibrosis with severe bridging fibrosis, and the possibility of forming cirrhosis. The score was set to 0–4 points in sequence. According to the degree of inflammation, it was divided into no infiltration of inflammatory cells, mild infiltration of inflammatory cells (less than 2 inflammatory cells/20× magnification), moderate infiltration of inflammatory cells (2–4 inflammatory cells/20× magnification), and severe infiltration of inflammatory cells (more than 4 inflammatory cells/20× magnification). The score was set to 0–3 points in sequence. According to bile duct injury, it could be divided into slight bile duct injury, mild bile duct injury, and obvious bile duct injury, with a score of 0–2 points set in sequence. The higher the score, the more severe the tissue damage.

#### 2.7.3. Analysis of Expression Levels of Inflammatory Factor mRNA in Mouse Liver Tissue

Total RNA was extracted from mouse liver tissues using a reverse transcription kit, according to the manufacturer’s instructions. Total RNA was subjected to gel electrophoresis combined with A_260_/A_280_ and A_260_/A_230_ analyses for quality control. The cDNA was obtained by reverse transcription and stored at −20 °C for later use. We used qRT-PCR to analyze the inflammatory cytokines Interleukin-1 (*IL*-1) β, Interleukin-6 (*IL*-6), and tumor necrosis factor-α, (*TNF*-α), and *β-actin* was used as an internal reference gene and was synthesized by Biotechnology Co., Ltd. (Shanghai, China) using fluorescent quantitative primer sequences (Table 3). A Roche LC96 qRT-PCR instrument was used for analysis, and the quality of the qRT-PCR reaction was evaluated by setting the extension and dissolution curve conditions. The reaction conditions were as follows: pre-denaturation at 95 °C for 300 s; denaturation at 95 °C for 15 s; annealing at 61 °C for 20 s; and elongation at 72 °C for 30 s. Dissolution conditions were 65 °C for 30 s and 95 °C for 15 s. Cooling conditions were 37 °C for 30 s. The reaction was run for 35 cycles, the experiment was repeated 3 times, and the mRNA expression level was expressed using 2^−ΔΔCt^ [24].

### 2.8. Data Statistics

All experimental data were analyzed by one-way analysis of variance (ANOVA) using the IBM SPSS software (version 13.0). The data are expressed as mean ± standard deviation ($\bar{x}$ ± s) from the triplicate independent measurements, and *p* < 0.05 is considered to be statistically significant.

## 3. Results and Discussion

### 3.1. Single Factor Experiment

As presented in Figure 1A, when the solid–liquid ratio was within the range of 1:10–1:30, the total flavonoid yield of *S. bigelovii* increased with the increase in the solid–liquid ratio, and the total flavonoid content in its extract also gradually increased. The maximum flavonoid yield is reached when the solid–liquid ratio is 1:30, with a flavonoid yield of 4.75 ± 0.18 mg/g. When the solid–liquid ratio is greater than 1:30, the total flavonoid yield gradually decreases. This may be due to an increase in the material-to-liquid ratio that leads to an increase in the solvent osmotic pressure and the dissolution of other impurities, ultimately resulting in a decrease in the extraction rate of the target product. A small material-to-liquid ratio can lead to the incomplete extraction of total flavonoids. Therefore, a material-to-liquid ratio of 1:30 was selected as the appropriate condition. As presented in Figure 1B, when the ethanol concentration is between 40% and 60%, the total flavonoid yield exhibits an upward trend and reaches a peak of 4.60 ± 0.14 mg/g at 60%. When the ethanol concentration was between 60% and 80%, the total flavonoid yield slowly decreased, and this may be due to the increase in ethanol concentration, leading to an increase in alcohol-soluble substances and a decrease in the proportion of total flavonoids. Therefore, 60% ethanol was selected as the optimal solution. As presented in Figure 1C, the total flavonoid content first increases within the ultrasound temperature range of 30–70 °C, reaching a peak of 5.12 ± 0.28 mg/g at 50 °C, and then gradually decreases. It is possible that the internal structure of flavonoids is disrupted by the increase in temperature or that the increase in temperature leads to an increase in cell solutes, thus resulting in an increase in the viscosity of the system solution and affecting the dissolution of flavonoids. Therefore, 50 °C was selected as the appropriate condition. From Figure 1D, it can be observed that within the ultrasound power range of 100–300 W, the total flavonoid yield gradually increased at 100–250 W and reached a maximum value of 4.65 ± 0.23 mg/g at 250 W. The flavonoid yield gradually decreased, and this may have been due to the increase in ultrasound power, causing damage to the flavonoid structure. Therefore, an ultrasound power of 250 W was selected as the optimal condition.

### 3.2. Box–Behnken Response Surface Test Results

Using the total flavonoid yield in *S. bigelovii* as the response value, the experimental results were fitted and regressed using the Design Expert 10.0.4 software. The regression equation was obtained as Y = 5.66 + 0.083A − 0.076B − 0.027C − 0.12D + 0.17AB − 0.19AC + 0.30AD − 0.16BC − 0.035BD + 0.027CD − 0.54A^2^ − 0.62B^2^ − 0.50C^2^ − 0.32D^2^. The significance level of the model was *p* < 0.0001, and the difference was extremely significant, thus indicating that the model possessed a good fitting ability. The mismatch error value was *p* = 0.4382 > 0.05, the difference in mismatch terms was not significant, the experimental error was relatively small, and the interference that was received was small. Simultaneously, AC, AD, A^2^, B^2^, C^2^, and D^2^ (Table 4; *p* < 0 0001) exerted a highly significant effect on the yield of total flavonoids. The order of influence of each factor on yield was A > B > C. Among them, R^2^ = 0.9590 and R^2^_Adj_ = 0.9180 were both close to one, thus indicating that the model possessed a good degree of fit, and also that the model has good repeatability [25]. Therefore, this model could be used to predict and analyze the extraction process conditions of total flavonoids from *S. bigelovii*. Exploring and optimizing can lead to response surfaces that may mislead results, thus ensuring the quality of the model was crucial [26]. 

The lack of fit (LOF) F-test describes how the data change around the fitted model. If the model does not fit the data well, the F-number will be significant. A large *p*-value for the lack of fit was 0.4382 (*p* > 0.05) in Table 4 (PLOF), indicating that the F-statistic was not significant, and implying there was a significant model correlation between the variable and the process response. The R^2^ coefficient provides the proportion of total variation in the model’s predicted response, representing the ratio of the sum of squares of regression (SSR) to the total sum of squares (SST). It was desirable to have a high R^2^ value that was close to one and reasonably consistent with the adjusted R^2^. The higher R^2^ coefficient ensures the satisfactory adjustment of the quadratic model to the experimental data.

### 3.3. Response Surface Graphic Analysis and Determination of Optimal Extraction Process Conditions

The interaction structures of these two factors are provided in Figure 2. As presented in Figure 2A, the center of the contour map was elliptical, and the curve was steep, thus indicating that the interaction between the material-to-liquid ratio of factor A and the ethanol concentration of factor B was more significant. Additionally, the interaction effect of the contour map was significant (*p* = 0.0203). As presented in Figure 2B, the center of the contour map is elliptical, while the opening of the 3D map is downward, and the curve is steep. This exerted a significant impact on the extraction rate of total flavonoids (*p* = 0.0077), thus indicating that the interaction between factor A, the material-to-liquid ratio, and factor C (the ultrasound temperature) was more significant. From Figure 2C, it can be observed that the center of the contour map is elliptical, whereas the opening of the 3D map is downward and convex, thus indicating a significant interaction between the material-to-liquid ratio of Factor A and the ultrasonic power D (*p* = 0.0010). From Figure 2D, it can be observed that the center of the contour map is elliptical, and the curve is steep, thus indicating that the interaction effect of factors B (ethanol concentration) and C (ultrasound temperature) is more significant (*p* = 0.0293). As presented in Figure 2E,F, the contour map exhibited a smooth surface and did not significantly affect the extraction rate. Overall, the interaction between the solid–liquid ratio and the ultrasound power was the strongest and exerted a significant impact on the extraction rate of total flavonoids from *S. bigelovii*. The interaction between the ethanol concentration and the extraction temperature was slightly weaker but still exerted a significant impact on the extraction rate of total flavonoids from *S. bigelovii*. The optimal theoretical values of various parameters designed through Design Expert 10.0.4 software are as follows: a solid–liquid ratio of 1:34.14 (g/mL); an ethanol concentration of 59.51%; an extraction temperature of 49.71 °C; and an ultrasound power of 241.63W. The predicted yield is 5.67 ± 0.34 mg/g. In actual operation, we adjusted the above conditions to a material-to-liquid ratio of 1:30 (g/mL), an ethanol concentration of 60%, an extraction temperature of 50 °C, and an ultrasonic power of 250 W. The yield of *S. bigelovii* measured in the validation experiment was 5.71 ± 0.28 mg/g, and this is in good agreement with the theoretical value, thus indicating that the response surface model of this method possesses a certain degree of reliability.

Response surface methodology is a design optimization and analysis method that is based on multiple factor functions and intuitively expresses the predictive function model in a three-dimensional surface. It consists of a set of numerical analysis methods and mathematical statistics methods, which can be used to determine the impact of various factors and their interactions on non-independent variables in various processes. It can accurately express the relationship between factors and response values. Meanwhile, the response surface methodology can also utilize software optimization processes to achieve the fast and clear optimization of experimental methods [27]. The response surface methodology is widely used in disciplines such as chemistry, food, biology, and ecological environment, but it also has certain shortcomings. For example, the prerequisite for the response surface methodology is that the designed experimental points should include the best experimental conditions, and if the experimental points are not selected properly, the use of response surface methodology cannot obtain good optimization results. Before use, it is necessary to establish reasonable experimental factors and levels. The response surface method has limited ability in exploring nonlinear, highly interactive, or complex relationships and may not be able to capture all influencing factors and patterns of change. The effectiveness of the response surface method depends on the collected experimental data, and if the data quality is not high or there are insufficient data points, the accuracy and reliability of the model may be affected. Therefore, the response surface method is a useful tool that can help explain the relationships between variables and optimize the behavior of the dependent variable, but it is necessary to pay attention to its modeling assumptions and limitations, and apply them reasonably based on actual situations [28]. 

This article uses the response surface methodology to optimize the extraction process of total flavonoids from *S. bigelovii* and uses the ultrasound-assisted extraction method for extraction. Compared with traditional extraction methods, it could significantly increase the content of total flavonoids in *S. bigelovii*. The study on the extraction of total flavonoids from *S. bigelovii* showed that the total flavonoid content extracted by traditional methods was around 3.0% [29], while the response surface methodology optimized the extraction process to increase the flavonoid content to 5.7%, which was close to twice the traditional extraction process. The response surface optimization method could find the optimal process parameters for extracting total flavonoids from *S. bigelovii*, and combined with the ultrasound-assisted extraction technology, it could maximize the extraction of total flavonoids.

### 3.4. The Effects of Total Flavonoids from S. bigelovii on Serum ALT, AST, SOD, GSH-Px, and MDA in Mice with Acute Alcoholic Liver Injury

The results of the measurements of ALT, AST, SOD, GSH-Px, and MDA in the serum of mice with acute alcoholic liver injury caused by *S. bigelovii* are presented in Table 5. Compared to those of the blank control group, the levels of ALT and AST in the model group were significantly increased (*p* < 0.01) at 115.21 ± 13.56 U·L^−1^ and 112.18 ± 9.41 U·L^−1^, respectively. The activities of SOD and GSH-Px oxidase were significantly reduced (*p* < 0.01) at 202.63 ± 32.14 U/mg pro and 270.23 ± 64.64 U/mg pro, respectively. Concurrently, the content of MDA in the serum was significantly increased to 15.72 ± 2.26 μmol/mg pro, thus indicating the occurrence of acute alcoholic liver injury symptoms in mice. Compared to levels in the model group, the positive control group and the high dose of total flavonoids from *S. bigelovii* group can significantly (*p* < 0.01) reduce the levels of ALT (51.81 ± 9.89 U·L^−1^) and AST (44.34 ± 9.74 U·L^−1^) in mice. Indicating that *S. bigelovii* can reduce the damage to liver cells by lowering the levels of ALT and AST in serum [30]. The harmful substance MDA (7.54 ± 0.96 μmol/mg pro) was significantly reduced (*p* < 0.01), but the SOD (340.67 ± 26.52 U/mg pro) and GSH-Px (481.42 ± 31.06 U/mg pro) enzyme activity were increased in a dose-dependent manner. This result indicates that while reducing the content of the harmful substance MDA, *S. bigelovii* increases the activity or regeneration of SOD and GSH-Px to enhance the ability of the body to clear free radicals and reduce oxidative stress damage, thus achieving a protective effect on the liver. The activity of SOD and GSH-Px reflects the ability to scavenge reactive oxygen species, while the content of MDA reflects the level of lipid peroxidation. Under normal conditions, there is a balance between the antioxidant system and the prooxidative system in the body. After consuming alcohol, this balance is disrupted, resulting in a decrease in the activity of SOD and GSH-Px and an increase in the MDA content. Therefore, protecting the activity of the antioxidant system in the body and clearing free radicals in the body are of great significance in preventing and treating damage caused by oxidative stress [31].

### 3.5. The Effect of Total Flavonoids from S. bigelovii on Liver Tissue Pathology in Mice with Alcoholic Liver Injury

The H&E staining results of pathological sections of liver tissue from mice with acute alcoholic liver injury were analyzed and are presented in Figure 3A. Light microscopy revealed that the liver cell structure in the blank control group was clear, the nucleus was intact, and the liver cord was radially and neatly arranged. As presented in Figure 3B, the liver cells of the model group mice exhibited severe lesions with obvious cell edema and nuclear voids, and some liver cells showing patchy necrosis, inflammatory cell infiltration, and the disappearance of liver sinusoids, thus indicating the successful modeling of alcoholic liver injury in mice. Figure 3C presents the positive control group. Compared to the model group, the edema of liver cells was significantly reduced, the arrangement of liver cells was relatively neat, the nucleus was in good condition, and the liver cord exhibited a radial shape. As presented in Figure 3D–F, the morphology of the *S. bigelovii* group was compared to that of the model group, and the results revealed that the liver tissue damage in the *S. bigelovii* total flavonoid group gradually decreased with an increase in drug concentration. The edema of the liver cells was gradually alleviated, the liver cord gradually returned to a radial shape, the nucleus gradually became complete and clear, and the inflammatory cells disappeared [32]. The results of the Ishak evaluation of liver pathological changes are shown in Table 6. Compared with the control group, the MG group (*p* < 0.01) and the LSG group (*p* < 0.05) had more severe liver tissue damage. Compared with the MG group, the MSG group (*p* < 0.05) and the HSG group (*p* < 0.01) significantly reduced the degree of liver cell damage. These results indicate that the total flavonoids of *S. bigelovii* had the effects of protecting liver tissue and reducing liver injury.

### 3.6. The Effect of Total Flavonoids from S. bigelovii on the mRNA Expression of Inflammatory Factors in Liver Tissue of Mice with Alcoholic Liver Injury

This study investigated the levels of the inflammatory cytokines, *IL*-1 β, *IL*-6, and *TNF*-α, using quantitative analysis. As presented in Figure 4, compared to levels in the blank group, the mRNA expression levels of the three inflammatory factors in the model group were significantly upregulated, thus indicating that the model group may exhibit severe inflammatory reactions. Compared to the model group, both the positive control group and the high-dose group were able to significantly reduce the mRNA expression levels of the inflammatory factors *IL*-1 β, *IL*-6, and *TNF*-α (*p* < 0.01). In alcoholic liver injury, ethanol produces pro-inflammatory factors, *IL*-1, *IL*-6, and TNF, through Toll-like receptors (such as TLR4)-α, ultimately leading to inflammation and hepatocyte apoptosis [33]. Studies have shown that the levels of *IL*-1 in patients with alcoholic liver cirrhosis significantly increase [34]. The results of this study indicate that *S. bigelovii* can significantly inhibit the expression of inflammatory factors, and this indicates that *S. bigelovii* protects the liver by increasing anti-inflammatory activity.

## 4. Conclusions

Response surface analysis was used to determine the optimal conditions for the microwave-assisted extraction of total flavonoids from *S. bigelovii*. The extraction process conditions for total flavonoids from *S. bigelovii* were determined to be a solid–liquid ratio of 1:30 (g/mL), an ethanol concentration of 60%, an extraction temperature of 50 °C, and an ultrasound power of 250 W. The yield was 5.71 ± 0.28 mg/g, and this is close to the predicted value of 5.67 ± 0.34 mg/g. This indicated that the extraction process was relatively stable and highly reliable. The research results from acute alcoholic liver injury indicate that the total flavonoids of *S. bigelovii* can inhibit the levels of ALT and AST in serum, reduce MDA levels, increase the activity of the antioxidant enzymes SOD and GSH-Px, and inhibit the expression of inflammatory factors. Therefore, the hepatoprotective effect of total flavonoids from *S. bigelovii* primarily reduces the expression levels of inflammatory factors in the body to inhibit the occurrence of inflammation and reduce the production of ROS, thus reducing oxidative stress damage. This study indicates that the flavonoids of *S. bigelovii* exert a significant protective effect against alcoholic liver injury and support the specific clinical application of *S. bigelovii* in the treatment of alcoholic liver disease.

## Figures and Tables

**Figure 1 foods-13-00647-f001:**
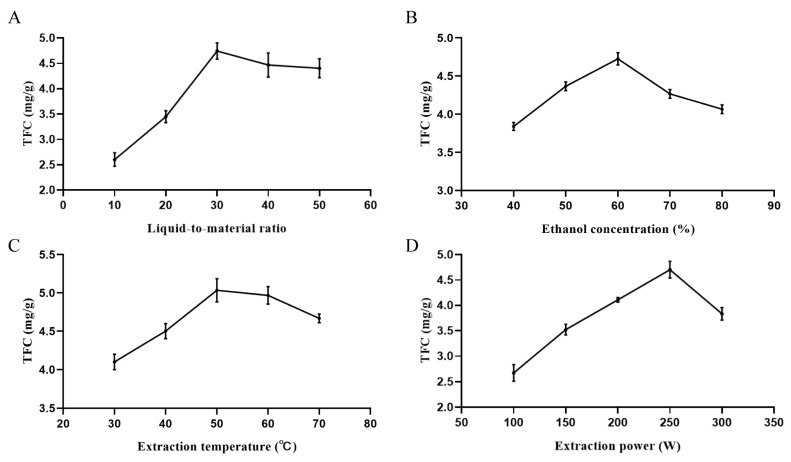
The influence of different extraction conditions on the yield of total flavonoids: (**A**) liquid-to-material ratio; (**B**) ethanol concentration; (**C**) extraction temperature; and (**D**) extraction power.

**Figure 2 foods-13-00647-f002:**
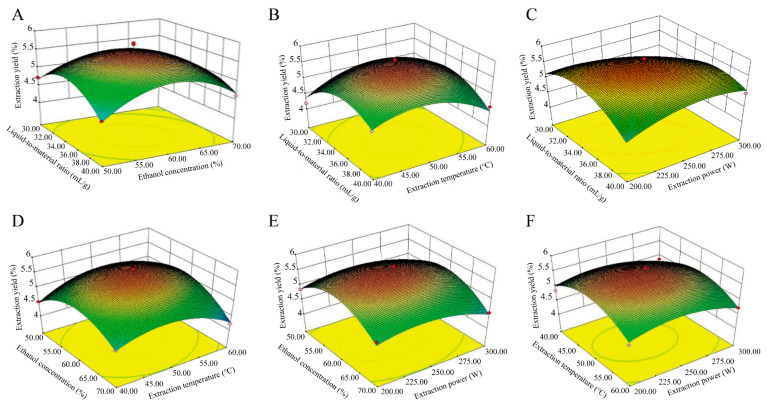
Response surface graph of interaction between different extraction conditions: (**A**) interaction between liquid-to-material ratio and ethanol concentration; (**B**) interaction between liquid-to-material ratio and extraction temperature; (**C**) interaction between liquid-to-material ratio and extraction power; (**D**) interaction between ethanol concentration and extraction temperature; (**E**) interaction between ethanol concentration and extraction power; and (**F**) interaction between extraction temperature and extraction power.

**Figure 3 foods-13-00647-f003:**
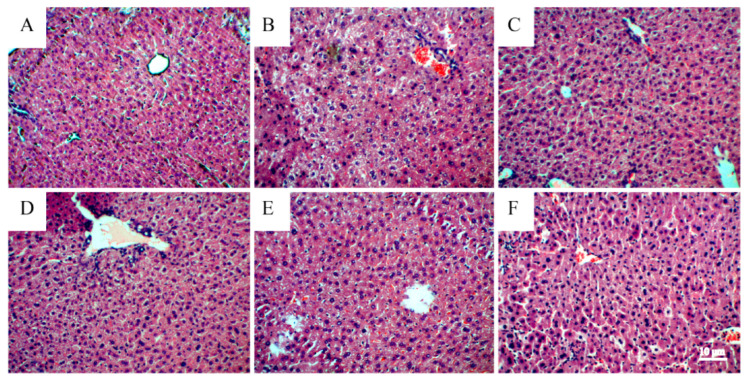
Analysis of liver tissue slices from different treatment groups: (**A**) control group, (**B**) model group (0.9% physiological saline), (**C**) positive group (silymarin, 200 mg/kg), (**D**) low *S. bigelovii* group (100 mg/kg), (**E**) middle *S. bigelovii* group (200 mg/kg), and (**F**) high *S. bigelovii* group (400 mg/kg). The tissue sections (5 µm) were stained with H&E (×200).

**Figure 4 foods-13-00647-f004:**
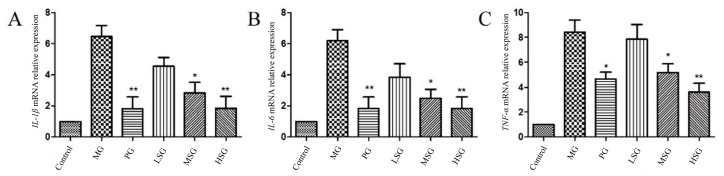
Results of relative expression levels of inflammatory factors, *IL*-1 β, *IL*-6, and *TNF*-α, in different treatment groups: (**A**) *IL*-1 β relative expression, (**B**) *IL*-6 relative expression, and (**C**) *TNF*-α relative expression. Control: control group; MG: model group (0.9% physiological saline); PG: positive group (silymarin; 200 mg/kg); LSG: low *S. bigelovii* group (100 mg/kg); MSG: middle *S. bigelovii* group (200 mg/kg); and HSG: high *S. bigelovii* group (400 mg/kg). Compared to the model group, * *p* < 0.05, ** *p* < 0.01.

**Table 1 foods-13-00647-t001:** The level of factors for the experiment.

Factors	Low	Center	High
Liquid to material ratio (mL/g, A)	−1 (25)	0 (30)	1 (35)
Ethanol concentration (%, B)	−1 (50)	0 (60)	1 (70)
Extraction temperature (°C, C)	−1 (40)	0 (50)	1 (60)
Extraction power (W, D)	−1 (200)	0 (250)	1 (300)

**Table 2 foods-13-00647-t002:** Box–Behnken design and observed responses.

Run	Independent Variable				Response (Flavonoids Extraction Rate (%))
	A	B	C	D	
	Liquid-to-Material Ratio (*v*/*v*%)	Ethanol Concentration (%)	Extraction Temperature (°C)	Extraction Power (W)	
1	25	50	50	250	4.72
2	35	50	50	250	4.50
3	25	70	50	250	4.14
4	35	70	50	250	4.61
5	30	60	40	200	4.84
6	30	60	60	200	4.78
7	30	60	40	300	4.87
8	30	60	60	300	4.72
9	25	60	50	200	4.62
10	35	60	50	200	4.50
11	25	60	50	300	4.24
12	35	60	50	300	4.95
13	30	50	40	250	4.51
14	30	70	40	250	4.65
15	30	50	60	250	4.72
16	30	70	60	250	4.22
17	25	60	40	250	4.19
18	35	60	40	250	4.87
19	25	60	60	250	4.75
20	35	60	60	250	4.60
21	30	50	50	200	4.86
22	30	70	50	200	4.89
23	30	50	50	300	4.65
24	30	70	50	300	4.54
25	30	60	50	250	5.61
26	30	60	50	250	5.63
27	30	60	50	250	5.79
28	30	60	50	250	5.61
29	30	60	50	250	5.83

**Table 3 foods-13-00647-t003:** The RT-PCR primers used in this study.

Primer Name	Forward Primer Sequence (5′ to 3′)	Reverse Primer Sequence (5′ to 3′)
*β-actin*	GTCGTACCACAGGCATTGTGATGG	GCAATGCCTGGGTACATGGTGG
*IL*-1 β	CTCGTGCTGTCGGACCCAT	CAGGCTTGTGCTCTGCTTGTGA
*IL*-6	TTCCATCCAGTTGCCTTCTT	CAGAATTGCCATTGCACAAC
*TNF*-α	CTTCCAGAACTCCAGGCGGT	CACTTGGTGGTTTGCTACGACG

**Table 4 foods-13-00647-t004:** Estimated regression coefficient for the equation model and the analysis of variance (ANOVA) for the experimental model.

Parameter	Sun of Squares	Degree of Freedom	Mean Square Error	*F*-Value	*p*-Value	Significance
Model	5.69	14	0.41	23.39	<0.0001	***
A	0.078	1	0.078	4.48	0.0527	
B	0.069	1	0.069	3.97	0.0662	
C	9.677 × 10^−3^	1	9.677 × 10^−3^	0.56	0.4679	
D	0.15	1	0.15	8.57	0.0510	
AB	0.12	1	0.12	6.85	0.0203	*
AC	0.17	1	0.17	9.66	0.0077	**
AD	0.30	1	0.30	17.32	0.0010	***
BC	0.10	1	0.10	5.89	0.0293	*
BD	4.900 × 10^−3^	1	4.900 × 10^−3^	0.28	0.6038	
CD	3.720 × 10^−3^	1	3.720 × 10^−3^	0.21	0.6508	
A^2^	1.89	1	1.89	108.78	<0.0001	***
B^2^	2.40	1	2.40	137.90	<0.0001	***
C^2^	1.62	1	1.62	93.05	<0.0001	***
D^2^	0.67	1	0.67	38.67	<0.0001	***
Residual value	0.24	14	0.017			
Lack of fit	0.19	10	0.019	1.28	0.4382	
Pure error	0.058	2	0.014			
Sum	5.94	28				
R^2^	0.9590					
RAdj^2^	0.9180					

* *p* < 0.05, ** *p* < 0.01, *** *p* < 0.001.

**Table 5 foods-13-00647-t005:** The effects of *S. bigelovii* on ALT, AST, SOD, GSH-Px, and MDA in alcoholic liver injury mice ($\bar{x}$ ± s, *n* = 8).

Group	ALT (U·L^−1^)	AST (U·L^−1^)	SOD (U/mg pro)	GSH-Px (U/mg pro)	MDA (μmol/mg pro)
Control	27.36 ± 12.41	24.81 ± 1.75	389.24 ± 14.43	561.40 ± 42.36	5.12 ± 0.95
MG	115.21 ± 13.56 ^##^	112.18 ± 9.41 ^##^	202.63 ± 32.14 ^##^	270.23 ± 64.64 ^##^	15.72 ± 2.26 ^##^
PG	46.27 ± 5.23	40.32 ± 16.01	352.23 ± 13.57	502.23 ± 21.32	7.03 ± 1.42
LSG	88.17 ± 6.34	72.21 ± 11.35	251.25 ± 12.56	360.20 ± 51.22	13.18 ± 3.52
MSG	71.03 ± 13.45 *	66.92 ± 17.89 *	300.49 ± 21.04 *	455.11 ± 42.36 *	9.31 ± 1.89 *
HSG	51.81 ± 9.89 **	44.34 ± 9.74 **	340.67 ± 26.52 **	481.42 ± 31.06 **	7.54 ± 0.96 **

Control: control group; MG: model group (0.9% physiological saline); PG: positive group (silymarin; 200 mg/kg); LSG: low *S. bigelovii* group (100 mg/kg); MSG: middle *S. bigelovii* group (200 mg/kg); and HSG: high *S. bigelovii* group (400 mg/kg). Compared to the model group, * *p* < 0.05, ** *p* < 0.01, and compared to the control group, ^##^
*p* < 0.01.

**Table 6 foods-13-00647-t006:** Ishak assessment of the severity of pathological changes in liver tissue.

Evaluation Project	Control	MG	PG	LSG	MSG	HSG
Liver cell necrosis	0	2.67 ± 0.47	0.67 ± 0.24	1.80 ± 0.14	1.50 ± 0.53	0.80 ± 0.14
Liver fibrosis	0	1.75 ± 0.50	0	1.29 + 0.20	0	0
Inflammation level	0	3.25 ± 0.53	1.00 ± 0.00	2.16 ± 0.75	1.00 ± 0.00	0.60 ± 0.54
Bile duct injuries	0	1.25 ± 0.18	0	0.71 ± 0.48	0	0
Total score	0	8.08 ± 0.20 ^##^	1.67 ± 0.24	5.96 ± 0.39 ^#^	2.50 ± 0.27 *	1.40 ± 0.34 **

Control: control group; MG: model group (0.9% physiological saline); PG: positive group (silymarin; 200 mg/kg); LSG: low *S. bigelovii* group (100 mg/kg); MSG: middle *S. bigelovii* group (200 mg/kg); and HSG: high *S. bigelovii* group (400 mg/kg). Compared to the model group, * *p* < 0.05, ** *p* < 0.01, and compared to the control group, ^#^
*p* < 0.05, ^##^
*p* < 0.01.

## Data Availability

The original contributions presented in the study are included in the article, further inquiries can be directed to the corresponding author.

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
