# Peer review of "Optimization of Microwave-Assisted Extraction Process of Total Flavonoids from Salicornia bigelovii Torr. and Its Hepatoprotective Effect on Alcoholic Liver Injury Mice"

_foods, 2024, doi:10.3390/foods13050647_

Round 1
Reviewer 1 Report
Comments and Suggestions for Authors
The study aimed to determine optimal extraction conditions for total flavonoids from Salicornia bigelovii (S. bigelovii) using microwave-assisted extraction and assess their protective effects on alcoholic liver injury in mice. Results showed significant reductions in ALT and AST levels, decreased MDA levels, increased antioxidant enzyme activity, and inhibition of inflammatory gene expression. The study concludes that S. bigelovii flavonoids offer protection against alcoholic liver injury, providing valuable insights for potential high-value utilization and liver-protective drug development.
Some considerations should be taken into account by the authors:
line 161-162: The number of experiments (29 independent experiments) is mentioned, but there is no discussion on how this number was determined. Providing information on whether a power analysis or any statistical considerations were used to determine the sample size would be beneficial.
I recommend a more in-depth discussion of the potential limitations or assumptions associated with the response surface model. Additionally, while the agreement between predicted and measured values is highlighted, a discussion on the practical implications of the observed yield in terms of the overall quality or efficiency of the extraction process could further enhance the interpretation.
Semi-quantitative analysis of the histological changes will enhance the robustness of the analysis. While the description is detailed, incorporating quantitative measures, such as scoring systems for tissue damage or inflammatory cell infiltration, would enhance the objectivity of the evaluation. Please consult the following reference for more detailshttps://doi.org/10.1016/j.fct.2022.113198
Author Response
Reviewer #1:
- line 161-162: The number of experiments (29 independent experiments) is mentioned, but there is no discussion on how this number was determined. Providing information on whether a power analysis or any statistical considerations were used to determine the sample size would be beneficial.
Answer: The concerns from the reviewer were very important, we apologize for not specifying the number of response surface experiments clearly. In this study, the number of independent experiments was based on a single factor experiment, and according to the Box-Behnken central combination design principle, we selected four factors, such as liquid to material ratio, ethanol concentration, extraction temperature, and extraction power, which have a significant impact on the extraction rate of total flavonoids as independent variables, and the extraction rate of total flavonoids from S. bigelovii as the response value. Design a total of 29 experiments with four factors and three levels, of which 24 were factorial experiments and 5 were central experiments, which were used to estimate experimental errors. The factorial experiment mainly examines the influence of four factors on the target variable of total flavonoids in S. bigelovii. Under the conditions of influencing independent variables, there were some controllable and uncontrollable random factors. When there are many factors and the levels included in the factors were divided too finely, it will make the interaction content numerous, not only inconvenient to calculate, but also very complex to explain them in detail. Therefore, only selecting a representative combination of levels for experimentation was to ignore some insignificant interaction effects. The analysis of variance (ANOVA) method was the most important statistical method for analyzing multi factorial experiments. It operated on 29 sets of experiments in this study, and the experimental results were first tested for homogeneity of variance before conducting analysis of variance. We have made corresponding modifications in the manuscript and highlighted them in different colors.
- I recommend a more in-depth discussion of the potential limitations or assumptions associated with the response surface model. Additionally, while the agreement between predicted and measured values is highlighted, a discussion on the practical implications of the observed yield in terms of the overall quality or efficiency of the extraction process could further enhance the interpretation.
Answer: Thanks a lot for reminding us to improve the manuscript quality. We have added relevant discussion content and made modifications in the manuscript. The specific modifications are as follows:
Response surface methodology is a design optimization and analysis method that is based on multiple factor functions and intuitively expresses the predictive function model in a three-dimensional surface. It consists of a set of numerical analysis methods and mathematical statistics methods, which can be used to determine the impact of various factors and their interactions on non-independent variables in various process processes. It can accurately express the relationship between factors and response values. Meanwhile, response surface methodology can also utilize software optimization processes to achieve fast and clear optimization of experimental methods [1]. The response surface methodology is widely used in disciplines such as chemistry, food, biology, and ecological environment, but it also has certain shortcomings. For example, the prerequisite for the response surface methodology is that the designed experimental points should include the best experimental conditions, if the experimental points are not selected properly, the use of response surface methodology cannot obtain good optimization results. Before use, it is necessary to establish reasonable experimental factors and levels. The response surface method has limited ability in exploring nonlinear, highly interactive or complex relationships, and may not be able to capture all influencing factors and patterns of change. The effectiveness of the response surface method depends on the collected experimental data, if the data quality is not high or there are insufficient data points, the accuracy and reliability of the model may be affected. Therefore, the response surface method is a useful tool that can help explain the relationships between variables and optimize the behavior of the dependent variable, but it is necessary to pay attention to its modeling assumptions and limitations, and apply them reasonably based on actual situations [2].
This article uses response surface methodology to optimize the extraction process of total flavonoids from S. bigelovii, and uses ultrasound assisted extraction method for extraction. Compared with traditional extraction methods, it could significantly increase the content of total flavonoids in S. bigelovii. The study on the extraction of total flavonoids from S. bigelovii, showed that the total flavonoid content extracted by traditional methods was around 3.0% [3], while the response surface methodology optimized the extraction process to increase the flavonoid content to 5.7%, which was close to twice the traditional extraction process. The response surface optimization method could find the optimal process parameters for extracting total flavonoids from S. bigelovii, and combined with ultrasound assisted extraction technology, it could maximize the extraction of total flavonoids.
References
- Breig, S. J. M.; Luti, K. J. K., Response surface methodology: A review on its applications and challenges in microbial cultures. Materials Today: Proceedings. 2021, 42, 2277-2284.
- Kumari, M.; Gupta, S. K., Response surface methodological (RSM) approach for optimizing the removal of trihalomethanes (THMs) and its precursor’s by surfactant modified magnetic nanoadsorbents (sMNP) - An endeavor to diminish probable cancer risk. Scientific Reports. 2019, 9, (1), 18339.
- Wang, D.; Wang, J.; Zheng, J.; Shang, Y.; Yu, X., Ca2+ and ABA on the Accumulation of GABA and Flavonoids in Germinated Salicornia bigelovii Torr. under NaCl Stress. Journal of Food and Nutrition Research. 2021, 9, (5), 263-273.
- 3. Semi-quantitative analysis of the histological changes will enhance the robustness of the analysis. While the description is detailed, incorporating quantitative measures, such as scoring systems for tissue damage or inflammatory cell infiltration, would enhance the objectivity of the evaluation.
Answer: The concerns from the reviewer were very important. We have conducted a semi-quantitative analysis of the histological changes base on the liver pathology scoring system. The specific content was as follows:
This study was based on the liver pathology scoring system, also known as the Ishak scoring system, to evaluate the degree of pathological changes in liver tissue. The main evaluation indicators include liver cell necrosis, fibrosis, inflammation, and biliary tract injury [1]. According to the degree of liver cell necrosis, it was divided into no necrosis, less than 50% necrosis, 50% -75% necrosis, and more than 75% necrosis, with a score of 0-3 points set in sequence. According to the degree of fibrosis, it could be divided into: no fibrosis, only the presence of fibrosis phenomenon but not severe, mild fibrosis had reticular fibrosis and interlobular fibrosis, moderate fibrosis had obvious coarse fibrosis in reticular and interlobular fibrosis, severe fibrosis had severe bridging fibrosis, and the possibility of forming cirrhosis. The score was set to 0-4 points in sequence. According to the degree of inflammation, it was divided into no infiltration of inflammatory cells, mild infiltration of inflammatory cells (less than 2 inflammatory cells / 20 X magnification), moderate infiltration of inflammatory cells (2-4 inflammatory cells / 20 X magnification), and severe infiltration of inflammatory cells (more than 4 inflammatory cells / 20 X magnification). The score was set to 0-3 points in sequence. According to bile duct injury, it could be divided into slight bile duct injury, mild bile duct injury, and obvious bile duct injury, with a score of 0-2 points set in sequence. The higher the score, the more severe the tissue damage. The results of Ishak evaluation of liver pathological changes were shown in the table 1. Compared with the control group, the MG group (P<0.01) and the LSG (P<0.05) had more severe liver tissue damage. Compared with the MG group, the MSG group (P<0.05) and the HSG group (P<0.01) significantly reduced the degree of liver cell damage. This indicates that the total flavonoids of S.bigelovii had the effect of protecting liver tissue and reducing liver injury. We have made corresponding modifications in the manuscript and highlighted them in different colors.
Table 1 Ishak assessment of the severity of pathological changes in liver tissue
Evaluation project |
Control |
MG |
PG |
LSG |
MSG |
HSG |
Liver cell necrosis |
0 |
2.67±0.47 |
0.67±0.24 |
1.80±0.14 |
1.50±0.53 |
0.80±0.14 |
Liver fibrosis |
0 |
1.75±0.50 |
0 |
1.29+0.20 |
0 |
0 |
Inflammation level |
0 |
3.25±0.53 |
1.00±0.00 |
2.16±0.75 |
1.00±0.00 |
0.60±0.54 |
Bile duct injuries |
0 |
1.25±0.18 |
0 |
0.71±0.48 |
0 |
0 |
Total score |
0 |
8.08±0.20## |
1.67±0.24 |
5.96±0.39# |
2.50±0.27* |
1.40±0.34** |
Control: control group, MG: model group (0.9% physiological saline), PG: positive group (silymarin, 200 mg/kg), LSG: low S.bigelovii group (100 mg/kg), MSG: middle S.bigelovii group (200 mg/kg), HSG: high S.bigelovii group (400 mg/kg).Compared to the model group *P<0.05,**P<0.01, compared to the control group #P<0.05,##P<0.01.
References
- Gibson-Corley, K. N.; Olivier, A. K.; Meyerholz, D. K., Principles for Valid Histopathologic Scoring in Research. Veterinary Pathology. 2013, 50, (6), 1007-1015.

Reviewer 2 Report
Comments and Suggestions for Authors
General appreciation
The manuscript article entitled: Optimization of the Microwave-Assisted Extraction Process of Total Flavonoids from Salicornia bigelovii Torr. and its hepatoprotective effect on mice with alcoholic liver damage, is topical since it is the protection of the liver, which is a vital organ in the body's metabolism. The article manuscript is very well written in clear and simple English.
The subject is interesting and could be improved:
The authors deal with an interesting theme, but unfortunately the problem is badly formulated. This introduction is very long and needs to be completely reworked for improvement. The authors should emphasize the usefulness of microwave-assisted extraction. Indeed, extraction is not sufficiently optimal with traditional methods (decoction, infusion, etc.).
We suggest deleting this section (lines 36-40) and dealing only with bioactive compounds, in particular flavonoids. You could fuss over it with paragraph 45-53
We suggest that this part (lines 55-68) be deleted as it is not essential. The authors should maximise efforts on the interest of microwave assisted extraction.
This part (line 83-85) is about methodology
Specify the identification number (lines 94)
This part is not covered in the methodology. The authors decided to use a three-dimensional model (response surface), it would be contradictory to use a one-factor model.
An analysis of variance (ANOVA) is often accompanied by a model-validation statistic called a lack of fit (LOF) test. A statistically significant LOF test often worries experimenters because it indicates that the model does not fit the data well.
When there is significant lack of fit, check how the replicates were run!!! If the replicates have been run correctly, then the significant LOF indicates that perhaps the model is not fitting all the design points well. Consider transformations (check the Box Cox diagnostic plot). Check for outliers. It may be that a higher-order model would fit the data better. In that case, the design probably needs to be augmented with more runs to estimate the additional terms. Alternatively, to improve the fit of the model, it may be necessary to use the model as is and rely on confirmatory tests to validate the experimental results. In this case, it is important to be aware of the possibility that the model may not be a very good predictor of the process in specific areas of the design space. Please provide us with clear information on lack of fit. The effectiveness of the adjustment depends not only on the R2 or R-adj but above all on the lack of fit.
Conclusion
Authors should take into account our observations which will contribute to improving the scientific quality of the manuscript article.
Author Response
Reviewer #2:
- The authors deal with an interesting theme, but unfortunately the problem is badly formulated. This introduction is very long and needs to be completely reworked for improvement. The authors should emphasize the usefulness of microwave-assisted extraction. Indeed, extraction is not sufficiently optimal with traditional methods (decoction, infusion, etc.).
Answer: Thank you very much for the reviewer's reminder. We have streamlined the introduction section and added relevant content on microwave assisted extraction, the specific modifications are as follows:
The traditional extraction methods for flavonoids include solvent extraction, crystallization, microfluidic technology, vacuum extraction, Soxhlet extraction and so on. However, these methods have problems such as long extraction time, expensive extraction equipment, low extraction efficiency, and the need for high-purity solvents. The ultrasonic assisted extraction technology is based on the presence, polarity, solubility, and other active ingredients in the substance, which quickly enter the solvent under the action of ultrasonic waves to obtain a multi-component mixture, and then obtain the active substance monomer through appropriate separation and purification techniques. At present, this method has been widely used for the extraction and separation of effective components in natural products. Combining ultrasonic waves with traditional solvent extraction has the advantages of reducing extraction time, targeted heating, reducing solvent consumption, and high extraction rate, making it an effective method for extracting total flavonoids [1]. It has been used to extract various plant flavonoids [2,3]. Research has found that using ultrasound assisted extraction of total flavonoids can shorten the extraction time and improve the yield compared to traditional Soxhlet extraction methods [4]. Ultrasound assisted extraction avoids the damage of high temperature to active ingredients, and the main influencing factors of ultrasound extraction include ultrasound frequency, extraction time, etc. Therefore, finding the appropriate parameters in ultrasound assisted extraction is the key to improving extraction efficiency. Meanwhile, the combination of ultrasound technology and other emerging technologies will be a research hotspot. With the continuous updating of research technology, ultrasonic extraction technology will show more extensive application prospects in fields such as food, medicine, and chemical industry [5].
References
- Yusoff, I. M.; Mat Taher, Z.; Rahmat, Z.; Chua, L. S., A review of ultrasound-assisted extraction for plant bioactive compounds: Phenolics, flavonoids, thymols, saponins and proteins. Food Research International. 2022, 157, 111268.
- Chen, F.; Wang, B.; Zhao, G.; Liang, X.; Liu, S.; Liu, J., Optimization extraction of flavonoids from peony pods by response surface methodology, antioxidant activity and bioaccessibility in vitro. Journal of Food Measurement and Characterization. 2023.
- Zhao, T.; Ding, Y.; Sun, W.; Turghun, C.; Han, B., Ultrasonic-assisted extraction of flavonoids from Nitraria sibirica leaf using response surface methodology and their anti-proliferative activity on 3T3-L1 preadipocytes and antioxidant activities. Journal of food science. 2023, 88, (6), 2325-2338.
- Molina, G. A.; González-Fuentes, F.; Loske, A. M.; Fernández, F.; Estevez, M., Shock wave-assisted extraction of phenolic acids and flavonoids from Eysenhardtia polystachya heartwood: A novel method and its comparison with conventional methodologies. Ultrasonics Sonochemistry. 2019, 104809.
- Quang, N. V.; Anh, N. N.; Thu, Q. T. M.; Vn, T. T.; Cuong, D.; Tam, N. Q.; Thuy, T. T. T., Optimization of Ultrasound-assisted Extraction of Ulvan from Green Seaweed Ulva lactuca. VNU Journal of Science: Natural Sciences and Technology. 2022, 38, (3), 70-76.
- We suggest deleting this section (lines 36-40) and dealing only with bioactive compounds, in particular flavonoids. You could fuss over it with paragraph 45-53.
Answer: Thank you very much for the reviewer's feedback on the revisions. We have removed the content of lines 36-40, reducing redundant information and making the introduction concise and highlighting key points.
3.We suggest that this part (lines 55-68) be deleted as it is not essential. The authors should maximise efforts on the interest of microwave assisted extraction.
Answer: Thank you very much for the reviewer's suggestions. We have removed the content of lines 55-68 and focused on the ultrasound assisted extraction technology, as stated in the answer to the first question. We hope the results can meet the requirements of journals and reviewers.
- This part (line 83-85) is about methodology.
Answer: Thank you very much for the reviewer's suggestions, The content of line83-85 belongs to the research method, and we have provided a detailed description of the method used in this article in the research method section. Therefore, this section is a duplicate description, and we have deleted it as required to make the overall description of the article more concise and clearer.
- Specify the identification number (lines 94).
Answer: Thank you very much for the valuable comments provided by the reviewer. We have supplemented the batch number of the product to be purchased in lines94, to make the material information more complete. The modifications made in the manuscript have been marked in red.
- This part is not covered in the methodology. The authors decided to use a three-dimensional model (response surface), it would be contradictory to use a one-factor model.
Answer: Thank you very much for the reviewer's reminder. As we all know, the response surface analysis is a statistical experimental design method used to optimize the impact of multiple independent variables on a response variable. Before making a response surface, it is necessary to conduct a single factor analysis to determine the reasonable range of all factors (including the optimal value), and then select the optimal values of several factors with greater influence among the single factors. The values on both sides of the optimal value are used as the optimal interval for conducting response surface experiments to fit the best conditions and equations. Therefore, the purpose of using a single factor experiment in this manuscript is to determine the factors that affect the total flavonoids of S. bigelovii and their optimal values, providing a reasonable numerical range for response surface methodology experimental design.
- An analysis of variance (ANOVA) is often accompanied by a model-validation statistic called a lack of fit (LOF) test. A statistically significant LOF test often worries experimenters because it indicates that the model does not fit the data well.
Answer: Thank you very much for the valuable feedback provided by the reviewer, which will have a significant impact on improving the quality of the manuscript, we have added relevant analysis. The lack of fit (LOF) F-test describes how the data changes around the fitted model. If the model does not fit the data well, the F-number will be significant. Large P-values for lack of fit was 0.4382 (P>0.05) in Table 4 (PLOF), indicating that the F-statistic was not significant, implying there was a significant model correlation between the variable and the process response. The R2 coefficient gives the proportion of total variation in the model's predicted response, representing the ratio of the sum of squares of regression (SSR) to the total sum of squares (SST). It was desirable to have a high R2 value (0.9590) close to 1 and reasonably consistent with the adjusted R2 (0.9180). The higher R2 coefficient ensures the satisfactory adjustment of the quadratic model to the experimental data. The modifications made in the manuscript have been marked in red.
